# High Predatory Capacity of a Novel *Arthrobotrys oligospora* Variety on the Ovine Gastrointestinal Nematode *Haemonchus contortus* (Rhabditomorpha: Trichostrongylidae)

**DOI:** 10.3390/pathogens10070815

**Published:** 2021-06-29

**Authors:** Fabián Arroyo-Balán, Fidel Landeros-Jaime, Roberto González-Garduño, Cristiana Cazapal-Monteiro, Maria Sol Arias-Vázquez, Gabriela Aguilar-Tipacamú, Edgardo Ulises Esquivel-Naranjo, Juan Mosqueda

**Affiliations:** 1Immunology and Vaccines Laboratory, Facultad de Ciencias Naturales, Universidad Autónoma de Querétaro, Querétaro 76140, Mexico; 2CONACYT-Unidad Regional Universitario de Zonas Áridas, Universidad Autónoma Chapingo, Bermejillo 35230, Mexico; 3Laboratorio de Microbiología Molecular, Unidad de Microbiología Básica y Aplicada, Facultad de Ciencias Naturales, Universidad Autónoma de Querétaro, Querétaro 76140, Mexico; landeros@uaq.mx; 4Centro Regional Universitario Sursureste, Universidad Autónoma Chapingo, Teapa 86800, Mexico; robgardu@hotmail.com; 5COPAR (Control of Parasites), Animal Pathology Department, Veterinary Faculty, Santiago de Compostela University, Campus Universitario, 27002 Lugo, Spain; cristianafcm@gmail.com (C.C.-M.); mariasol.arias@usc.es (M.S.A.-V.); 6C.A. Salud Animal y Microbiologia Ambiental, Facultad de Ciencias Naturales, Universidad Autónoma de Querétaro, Av. de las Ciencias s/n Col Juriquilla, Querétaro 76230, Mexico; gabriela.aguilar@uaq.mx

**Keywords:** parasitosis, antihelmintic, biocontrol, nematofagous fungi, *Arthrobotrys oligospora*, light response

## Abstract

With the worldwide development of anthelmintic resistance, new alternative approaches for controlling gastrointestinal nematodes in sheep are urgently required. In this work, we identified and characterized native nematode-trapping fungi. We collected seven isolates of fungi with the capacity to form adhesive, three-dimensional networks as the main mechanism to capture, kill, and consume nematodes. The nematode-trapping fungi were classified into two groups; the first group includes the R2-13 strain, showing faster growth, abundant aerial hyphae, scarce conidia production, bigger conidia, and it formed a clade with *Arthrobotrys oligospora* sensu stricto. The second comprises the A6, A12, A13, R2-1, R2-6, and R2-14 strains, showing a growth adhering to the culture medium, forming little aerial hyphae, smaller conidia, and these formed a sister clade to *A. oligospora*. Except for the R2-6 strain, conidia production was induced by light. In all the strains, the predatory capacity against the sheep gastrointestinal nematode *Haemonchus contortus* was greater than 58% compared with the control group. The A6 and A13 strains were the most active against the infective *H. contortus* third instar (L_3_) larvae, with an average capture capacity of 91%. Altogether, our results support evidence for a novel *A. oligospora* variety with high nematode-trapping activity and promissory in helminthic control.

## 1. Introduction

The parasitic control programs based solely on the administration of pharmacological treatments for nematodes affecting the ovine industry in the tropics can be considered a temporary solution because they do not act on the free stages of the soil life cycle of the nematodes. Animals can reinfect themselves upon consuming forage contaminated with those stages, and the serious downsides to the use of agrochemical products are their high costs, the presence of resistance [1], and their possible harmful effect on the environment. To solve these problems, a drastic reduction in pharmacological treatments in the coming decades is needed. Therefore, new alternatives or supplementary methods are actively sought. In nature, there are numerous antagonists of helminths, such as fungi, bacteria, viruses, protozoa, and mites. Among them, nematophagous fungi likely play a role as natural enemies of parasitic nematodes of plants and animals because they can capture, parasitize, or paralyze some free-living nematodes. In a simplified fashion, nematophagous fungi have been traditionally classified into predators or endoparasites [2]. Nowadays, it seems more appropriate to classify them according to how they feed on nematodes and the mechanisms involved in these processes, thus finding fungi with ovicidal [3,4] or larvicidal properties [5].

Nematophagous fungi comprise over 200 species of taxonomically diverse fungi that can capture, kill, and digest nematodes [6]. Noticeably, most studies of nematophagous fungi have focused on predatory species belonging to the genera *Arthrobotrys*, *Duddingtonia*, and *Monacrosporium* as a biocontrol alternative for the use of anthelmintics in infective larvae of parasites in domestic animals [7]. This is because they possess some natural advantages, including a short life cycle, high reproductive ability, production of resistant spores, and survival in the saprophytic phase in the absence of hosts. Additionally, they are not pathogenic for humans, and they reduce parasitic populations rather than wiping them out, which maintains an important immunological stimulus in the animals [8,9]. Most of the studies on biological control of livestock nematodes are centered on *Duddingtonia flagrans* because it produces a large number of thick-walled chlamydospores that can be added to livestock feed as additives to control parasites, because when administered to animals they pass through the gastrointestinal tract and, when excreted into the environment, they colonize the fecal matter and capture and destroy a significant number of nematode larvae in situ [8,10].

Furthermore, some studies have documented the ability of *Arthrobotrys oligospora* to successfully feed on larvae in the third stage (L_3_) of cyathostomins that infect horses in Brazil [11] and some gastrointestinal nematodes (GIN) in sheep and cattle [12,13]. *Arthrobotrys* species were first described in Europe in 1850 by George Fresenius. *A. oligospora* was the first species of fungus known to actively capture nematodes [14] and is thought to be the most common nematode-trapping fungus and the most widespread. As a premise, *Arthrobotrys* species behave as generalist predators [15], indicating great potential in the biocontrol of parasitic nematodes.

This study aimed to isolate, identify, and evaluate in vitro the predatory ability of nematophagous fungus strains native from the Mexican state of Querétaro, and provide a first approach that can be used as biological control agents for infective larvae of ruminant gastrointestinal nematodes.

## 2. Materials and Methods

### 2.1. Gathering of Samples for Isolation (Area of Study)

The study was carried out from June 2018 through September 2018, in three locations in the Mexican state of Querétaro: one in El Marqués (20°42′37.5″ N, 100°15′24.6″ W) and two in Jalpan de Serra (21°11′31.7″ N, 99°27′23.0″ W) and (21°09159.1″ N, 99°21′01.7″ W). Samples of feces from sheep, cattle and horses were collected from the soil in depth ranges of 0 to 5 cm and in compost, pasture soil and under trees. In total, 55 samples were collected and stored in polyethylene bags, properly labeled, and transported to the facilities of the Microbiology Laboratory of the School of Natural Sciences of the Autonomous University of Querétaro for the isolation and identification of nematophagous fungi.

### 2.2. Infective Larvae of Haemochus Contortus

For obtaining infective larvae (L_3_) of *H. contortus,* we utilized fecal samples from a lamb previously infected under experimental conditions with L_3_ from *H. contortus* collected and cultivated for 10 days at room temperature [16,17]. Afterward, the larvae were collected using Baerman’s technique and quantified and stored at 4 °C before analysis [18].

### 2.3. Isolation of Nematophagous Fungi

Soil samples were processed with the sprinkled plate method outlined by Barron [2,17,19] for isolating nematophagous fungi. In this technique, for each soil sample, five subsamples of 0.5–1 g were collected and placed into 9 cm Petri dishes containing water-agar (WA). One milliliter of suspension containing 100–300 L_3_ of *H. contortus* was added using a micropipette. The plates were incubated at room temperature (23–26 °C), and from the fourth day onward, they were analyzed daily with an inverted microscope in search of hyphae, conidia, spores, or parasitized nematodes. Observations were maintained for a further week until 28 days after the incubation; the goal was to increase the likelihood of finding a fungal structure.

Once the presence of nematophagous fungi was confirmed, they were isolated to generate axenic cultures. In this regard, conidia, mycelia, or parasitized nematodes were collected by a mycological loop and then propagated on potato dextrose agar (PDA) medium with chloramphenicol (500 mg/L) to avoid the growth of bacteria. Samples were incubated at room temperature, and the growth of fungi was inspected 1 week later. Then, the colonies obtained were transferred into an antibiotic-free PDA medium for purification and replication.

### 2.4. Radial Growth

Mycelial plugs (0.5 cm) obtained from the periphery of the 72-hour-old colonies were transferred onto PDA or Vogel’s minimal medium with 2% sucrose (VMM). Three plates per strain were incubated under constant white light (0.586 µmol m^−2^ s^−1^) in a chamber equipped with two cool-white fluorescent tubes. The cultures were incubated at 27 °C for 72 h, photographed, and then the radial growth measured using the ImageJ software version 1.52a (http://imagej.nih.gov/ij/, accessed on 5 November 2018). The assay was carried out in triplicate.

### 2.5. Conidia Production

Mycelia plugs from 72-hour-old colonies were inoculated on PDA plates in triplicate and incubated at 27 °C for 10 days under constant white light (0.586 µmol m^−2^ s^−1^) or in darkness. Then, colonies were photographed, and conidia collected with 15 mL distilled water plus 0.1% Triton X-100 and counted in a Neubauer chamber. The assay was carried out in triplicate.

### 2.6. Morphological Identification

The taxonomical identification of nematophagous fungi was performed via morphological keys described by Cooke and Godfrey [19]. PDA blocks (approximately 1 cm^3^) were inoculated with mycelia of nematophagous fungi and incubated at 27 °C for 7 days. Conidiophores were analyzed at 10× and conidia at 40× magnification using a microscope Zeiss, Axio Scope.A1.

### 2.7. Conidial Measurements

Conidia were collected from colonies grown on PDA for 10 days in a chamber illuminated with white light as indicated above. A conidia suspension was prepared using 0.1% Triton X-100 solution, analyzed in a Zeiss, Axio Scope.A1 microscope and measured using the Image J software (version 1.52a). Total septum length, basal cell–septum length, and septum–distal cell length of conidia were measured as indicated in Appendix A and reported in ranges (minimum size–maximum size) with the average sizes in parentheses. Around 100 conidia per nematode-trapping strain collected from three different colonies were analyzed.

### 2.8. Molecular Characterization of the Isolated Fungi

#### 2.8.1. DNA Extraction

The strains were cultivated in a potato dextrose broth (PDB); the cultures were incubated in a chamber fitted with orbital stirrer running at 160 rpm for 72 h at 27 °C. Mycelia were collected by filtration, and later frozen in liquid nitrogen and ground to a fine powder in a mortar and pestle. The equivalent to 500 μL of powder mycelia were added to a 1.5 μL plastic tube with 500 μL of extraction buffer (7 M Urea, 0.35 M NaCl, 50 mM Tris pH 8.0, 20 mM EDTA, and 1% N-Lauryl Sarcosine). The mixture was vortexed for 2 min and incubated at room temperature for 20 min. Phenol extraction was performed two times as follows: 600 μL of phenol:chloroform (50 phenol:49 chloroform:1 isoamylic alcohol) was added, centrifuged at 10,000 rpm for 10 min, and the supernatant was transferred to new 1.5 mL tubes. One volume of isopropanol was added, homogenized, and centrifuged at 10,000 rpm for 10 min. The pellet was washed with 300 μL of 70% ethanol, vortexed for 2 min, and centrifuged at 10,000 rpm for 10 min. The pellet was dried at room temperature, resuspended in 100 μL of TE (10 mM Tris pH 8.0 and 1 mM EDTA) containing 0.1 UμL^−1^ RNase and incubated at 37 °C for 10 min before storage at −20 °C.

#### 2.8.2. Amplification of Ribosomal DNA

The ITS1-5.8S-ITS2 sequence was amplified using primers ITS1 (TCC GTA GGT GAA CCT GCG G) and ITS4 (TCC TCC GCT TAT TGA TAT GC) [20] from 100 ng of genomic DNA from each strain, following the recommendations of the manufacturer of Dream Taq DNA Polymerase (Thermo Fisher Scientific). The following thermocycling conditions were carried out: an initial denaturation step at 95 °C for 3 min followed by 40 cycles, each containing a denaturation step of 95 °C for 30 s, an alignment step at 58 °C for 30 s, and an extension step at 72 °C for 45 s. Reactions were stopped after a final extension step at 72 °C for 5 min. The amplicons were analyzed by agarose gel electrophoresis at 1%, and DNA fragments were purified using the QIAquick PCR Purification kit (QIAGEN) and sequenced using primer ITS1. The sequences were trimmed to select those with high quality, with well-defined peaks in the electropherograms using Chromas software, and they were deposited in the GenBank database of the Natural Center for Biotechnology Information (Table 1).

#### 2.8.3. Phylogenetic Analyses

The entire internal transcribed spacer region (ITS1-5.8S-ITS2) from nematode-trapping fungi isolated in this work and others previously published [6,21] (Table 1) were aligned to create a matrix (652 characters) using MacClade 4.08a software [22]. Then, it was analyzed using RAxML 7.2.6 software [23] to generate the maximum likelihood phylogenetic tree from 1000 replications using the GTRGAMMA model [24,25]. Branches were supported by bootstrap values obtained from 1000 replications. Only branches with support values above 50 were considered.

### 2.9. Evaluation of the Predatory Ability

Each fungus was cultured in water-agar Petri dishes (5 mm plugs) for 3 weeks at room temperature (25–28 °C), from 7-day-old pure colonies grown on PDA plates and obtained from the periphery of the culture. Afterward, 100 L_3_ of *H. contortus* were placed on each dish and were incubated for 5 days at room temperature. Ten replicates of each nematode-trapping isolate were carried out. As a control, ten replicates were carried out in parallel, depositing the larvae indicated above in Petri dishes containing water-agar without fungi. After 5 days, the surface of the Petri dishes was scraped with a spatula, washed with distilled water, and the liquid was collected in test tubes and refrigerated for 2 h at 4 °C to allow the larvae to move to the bottom of the tube. The volume of the tubes was reduced to 2 mL by means of a pipette, where the larvae were found. Ten aliquots of 20 μL were collected, and the retrieved larvae were counted under a stereoscopic microscope. To estimate the reduction percentages, the following formula was used [10]:$$\%\;Reduction=\frac{XC-XH}{XC}\times 100$$
where: *XC* = number of larvae retrieved from the control group; *XH* = number of larvae retrieved from the various strains of fungi.

### 2.10. Statistical Analysis

The retrieved larvae were counted, and a capture percentage was estimated. A variance analysis (ANOVA) was carried out with the statistical software SAS (SAS, 2004). Each isolated strain of predatory fungus was regarded as a treatment against the nematodes; ten specimens of each strain were grown in Petri dishes, and the results were processed with the formula given above, and the statistical difference of the capture was estimated among strains through the Duncan averages test with a confidence level of 95%.

## 3. Results

### 3.1. Isolation of Nematode-Trapping Fungi

Out of the 55 collected samples, 7 (12.7%) fungal isolates were recovered capable of capturing nematodes added to the water-agar medium using the sprinkled plate method. The A12 and R2-14 strains were obtained from 10 compost samples; the A13 and R2-6 strains were isolated from 15 samples of ovine feces; the R2-13 strain was isolated from 15 samples of bovine feces; the R2-1 strain was isolated from 2 common donkey feces samples; and the A6 strain was isolated from 2 samples of soil collected underneath a tree (Table 1). All seven isolates showed the presence of three-dimensional adhesive networks as the main device used to capture, kill, and digest nematodes (Figure 1), a widely documented characteristic described in species of the genus *Arthrobotrys* [6,15].

### 3.2. Nematode-Trapping Fungi Growth

The growth rate is a desirable characteristic for substrate or space competition that could be related to a major parasitic capacity. To characterize the isolated nematophagous strains, these were grown on PDA or VMM. All strains grew faster on VMM than on PDA medium (Figure 2), covering plates in around 3 and 4 days, respectively; however, denser and branched mycelia were produced on PDA, forming more aerial hyphae, conidiophores, and conidia. The R2-13 strain showed faster growth in both culture media, with a growth rate of 0.12 cm h^−1^ on VMM and 0.1026 cm h^−1^ on PDA while A6, A12, A13, R2-1, R2-6, and R2-14 strains showed a similar pattern, with growth rates of 0.1046, 0.0983, 0.0955, 0.1011, 0.1014, and 0.0992 cm h^−1^ on VMM, whereas in PDA they were 0.0875, 0.0864, 0.0809, 0.0894, 0.0959, and 0.086 cm h^−1^, respectively. These results indicate that at least two different groups of nematode-trapping fungi were isolated.

### 3.3. Nematode-Trapping Fungi Asexual Reproduction

Asexual spores (conidia) are the main inoculum used to propagate biocontrol agents. To analyze conidia production, strains were cultured on PDA at 27 °C for 10 days under light or darkness conditions. In light, the colonies were light peach in color and, except the R2-13 strain, showed a creased texture that was more pronounced under light. In the darkness, the colonies were not pigmented, except the R2-6 strain that was a light peach color under both conditions (Figure 3A). Furthermore, the R2-13 strain showed a fluffy colony morphology different from A6, A12, A13, R2-1, R2-6, and R2-14 strains that grew adhered to the substrate forming fewer aerial hyphae (Figure 3A or Figure 4A). Except for the R2-6 strain, conidia production was hardly observed in darkness, and it was induced by light. The A12 strain showed major conidia yield, producing an average of 13.11 × 10^6^ conidia per colony, whereas the R2-13 strain yield was the lowest, producing only 3.3 × 10^5^ conidia per colony (Figure 3B). Noticeably, the asexual reproduction of the R2-6 strain was independent of light (Figure 3C). Furthermore, although colony pigmentation and asexual reproduction were photoinduced, there was no relationship between colony color intensity and conidia yield, suggesting that mycelium and reproductive structures could be pigmented in nematode-trapping fungi. Our results suggest that those isolates correspond to at least two different kinds of nematode-trapping fungi.

### 3.4. Morphological Characterization of Nematode-Trapping Fungi

To better characterize the nematode-trapping strains, given that there were at least two different types of colonial morphology (Figure 3A and Figure 4 right column), microcultures were generated at 27 °C for 7 days. Conidiophores were formed from aerial septate hyphae, which were produced commonly among 3–9 conidia in a whorled pattern (Figure 4 middle column). Conidia is obovoid, plump, base apiculate, constricted at the septum, consist of two-celled, with a larger distal cell, and pear-shaped; the cells are of unequal size with the basal cell being smaller and nearer to the attachment point on the conidiophore (Figure 4, left column). These results suggest that those isolated strains show morphological characteristics and devices to trap nematodes typical to species of the genus *Artrobotrys*.

To carry out a deeper morphological characterization among isolates, we measured two-celled conidia as indicated in Appendix A, considering total length, septum, distal cell, and basal cell length. Noticeably, the R2-13 strain produced longer conidia (mean length, 21.4 μm; range, 16.0–30.3 μm), while the other six strains ranged from 17.5 to 18.8 μm on average (Table 2). Although the R2-13 septum length was slightly longer than the other isolates (range, 7.3–14.8 μm), the average was similar (8.2 μm), indicating that the R2-13 isolate produces slightly elongated-conidia (Table 2). Furthermore, R2-13 produced larger basal cells (range, 6.4–15.8 μm and 8.9 μm on average) than the A3, A12, A13, R2-1, R2-6, and R2-14 isolates, which ranged from 6.4 to 7.0 on average, indicating that the septum position is nearer to the conidia center as indicated by S-AC/BC-S ratio, where R2-13 was 1.39, while in the A6, A12, A13, R2-1, R2-6, and R2-14 isolates were 1.74, 1.79, 1.71, 1.78, 1.75, and 1.85, respectively. Altogether, our results clearly separate the R2-13 from the A6, A12, A13, R2-1, R2-6, and R2-14 isolates into at least two different nematode-trapping groups of the genus *Arthrobotrys*.

### 3.5. Phylogenetic Analyses of Arthrobotrys Isolates

To determine the identity of the isolates at the molecular level, the ITS1-5.8S-ITS2 sequences of nematophagous fungi were blasted against the GenBank database; all queries were highly similar to *Artrobotrys oligospora* rDNA sequences. To determine the genetic identity, the sequences were aligned to create a matrix and then a maximum likelihood phylogenetic tree was built using those previously documented (Table 1). The R2-13 strain formed a clade with *Arthrobotrys oligospora*; however, the A6, A12, A13, R2-1, R2-6, and R2-14 strains formed a sister clade to *Arthrobotrys oligospora* (Figure 5), where they differ by just over 2% in their sequences with respect to those previously published [6,21]. Considering the different morphological features described above and phylogenetic analyses, we propose that the A6, A12, A13, R2-1, R2-6, and R2-14 strains comprise a novel *Arthrobotrys oligospora var. queretana*.

### 3.6. High Depredatory Capacity of Nematophagous Native against H. Contortus In Vitro

The nematode *H. contortus* is the main sheep gastrointestinal nematode causing great losses in production systems. To explore the potential application, we analyze the depredatory capacity of nematode-trapping isolates against *H. contortus* in vitro. The statistical analysis showed differences between the different strains of *A. oligospora* (*p* < 0.05); and in Duncan’s separation of the means test, it was determined that strains A6 and A13 of the fungus *A. oligospora* were more active against larvae L_3_ of *H. contortus* of sheep with respect to the other isolates evaluated, with an average trapping capacity of 91%, the strains R2-1 and R2-13 of *A. oligospora* being those with the least predation, with a mean catch range of 58–64% (Figure 6).

## 4. Discussion

Nematophagous fungi are potential biocontrol agents to destroy L3 of GIN in domestic animals and are natural antagonists of nematodes. Larvicidal fungi capture nematodes by producing capturing organs in the vegetative mycelium. Li et al. [16] investigated phylogenies of nematophagous fungi deduced from sequence analyses of 5.8 S rDNA, 28 S rDNA, and β-tubulin genes, redefined the systematic classification of nematophagous fungi, and changed the morphological characterization of larvicidal fungi based on their types of capture organs. According to this revised generic concept, most of the species that produce adhesive networks in the genus *Monacrosporium* in previous literature were transferred to the genus *Arthrobotrys*.

The morphological features of the isolates in this study indicate that their nematode-trapping device is an adhesive tridimensional network that matches the morphology of the genus *Arthrobotrys* [6,15]. However, physiological and morphological characters, such as growth, mycelia, colonies, pigment, and conidia production, grouped our isolates into two clearly different kinds of nematode-trapping fungi. *Arthrobotrys* species have been taxonomically classified based on the conidiophore and two-celled conidia form, septum position, and measuring conidia width and length [19]. All seven isolates showed morphological characters, such as conidiophore not usually branched, septed, two-celled conidia in whorls along conidiophore, obovoide, plump, construited at the septum, base apiculate, and larger distal cell, as has been documented for *A. oligospora* [19], indicating that our isolates correspond to *A. oligospora*. Given that conidia were observed with different sizes among isolates, we measured conidia length, septum, basal cell, and distal cell. The R2-13 strain produces conidia and basal cells longer than the other six analyzed strains, and consequently, the septum position was nearer to the center in the conidia of the R2-13 strain, indicating that two different kinds of nematode-trapping fungi were isolated.

The R2-13 conidia length was 16.0–30.3 μm, whereas the lengths for A6, A12, A13, R2-1, R2-6, and R2-14 were 15.2–24.4, 15.6–21.6, 15.4–23.1, 15–21.9, 13.9–22.1, and 15.1–22.2, respectively. Cook and Godfrey documented a range of 22–33 μm for *A. oligospora*, which is similar to the R2-13 strain, but the other six strains produce smaller conidia. Another morphological difference was the S-DC/BC-S ratio of 1.39 for R2-13, whereas this value was between 1.71 and 1.85 for the other six strains, indicating that the R2-13 strain produces larger basal cells than the other six strains and, consequently, septum positioned nearer to the conidia center and supporting evidence for two different kinds of nematode-trapping fungi. In this regard, it was confirmed by maximum likelihood phylogenetic tree based on ITS sequences, demonstrating that R2-13 correspond to *A. oligospora* sensu stricto while the A6, A12, A13, R2-1, R2-6, and R2-14 isolates formed a sister clade, which we propose to name *A. oligospora var*. *queretana*.

The A6, A12, A13, R2-1, R2-6, and R2-14 ITS sequences differ by just over 2% in their sequences with respect to those previously published [6,21]. The average weighted infraspecific ITS variability for Ascomycota was calculated as 1.96% with a standard deviation of 3.73% [26]. Although we cannot discard the possibility that the A6, A12, A13, R2-1, R2-6, and R2-14 strains comprise a novel species of *Arthrobotrys*, we propose them as a novel variety of *Arthrobotrys oligospora*.

Although the strains were genetically close, they showed phenotypic differences. The R2-6 strain produced conidia in darkness and light, while the A3, A12, A13, R2-1, and R2-14 strains produced conidia in a strictly light-dependent manner. Consistently, the *A. oligospora* genome has protein homologs to Wc1/Blr1 (Id: 7017), Wc2/Blr2 (Id: 5819), and Vivid/Envoy1 (Id: 8452) [27], the main blue light photoreceptor widely described in the fungi kingdom, suggesting that conidia and pigment production are regulated in a similar manner as has been documented in fungi [28]. Interestingly, the R2-6 isolate produced conidia light-independently, representing a natural variant in the genetic circuit that governs asexual reproduction.

Furthermore, we evaluated the in vitro predatory activity of the seven native isolates of *A. oligospora* from Mexico on infective larvae of *H. contortus* of sheep. Our results show that these isolates could capture infective larvae, thus reducing the larva population by 58–91%. In this regard, *Arthrobotrys oligospora* isolates have high genetic diversity showing different responsiveness to ascarosides and predatory capacity [15], suggesting that our isolates respond differently to ascarosides produced by nematodes, inducing at a different level the development of tridimensional traps and representing in a different predatory capacity. Consistently, for larvae of strongyloids treated with *A. oligospora* for 14 days at 27 °C, the percentage of reduction of larvae was higher than 90% [29]. *A. oligospora* is the nematophagous fungus that is found in most parts of the world, which suggests that it is the one that best adapts to different environmental conditions, which is not the case for *D. flagrans*, which needs certain climatic characteristics for its development and nematophagous action. On the other hand, it is intended to be able to carry out tests in the future of the ways of applying it, for example: to elaborate concentrate feed that contain these spores and/or conidia. In the same way, formulated granules containing the fungal material can be prepared. This research work opens the door to future research on biological control using nematophagous fungi, characterizing new isolates that will also be explored for better action and efficacy. Our results indicate genetic diversity in our isolates representing a new *A. oligospora var*. *queretana* native from Queretaro State, México.

## 5. Conclusions

We isolated seven native strains from Querétaro State, México, which correspond to *Arthrobotrys oligospora* and a novel variety that formed a sister clade with *Arthrobotrys oligospora* sensu stricto. The A6 and A13 strains have high predatory capacity against *H. contortus*, a sheep gastrointestinal nematode.

## Figures and Tables

**Figure 1 pathogens-10-00815-f001:**
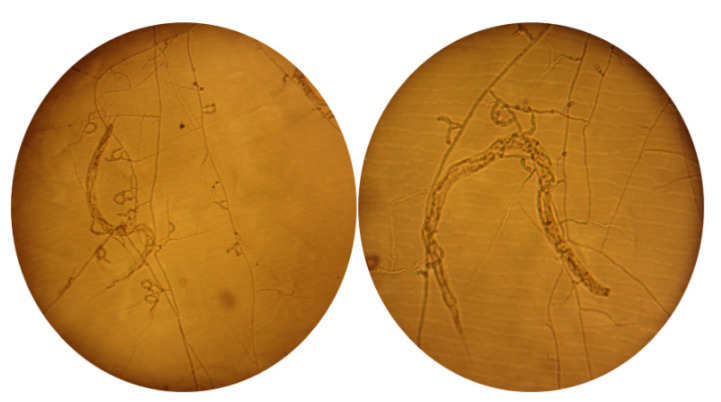
Trapped larvae of gastrointestinal nematodes in three-dimensional adhesive networks. The A6 strain was grown on water-agar for 7 days at 2 °C, an aqueous suspension with an undetermined number of *H. contortus* larvae was added, and they were incubated at room temperature for another 7 days. Right, nematode trapped by A6 strain three-dimensional network; the image on the left corresponds to a trapped nematode almost absorbed. The images were captured using a microscope at 10×.

**Figure 2 pathogens-10-00815-f002:**
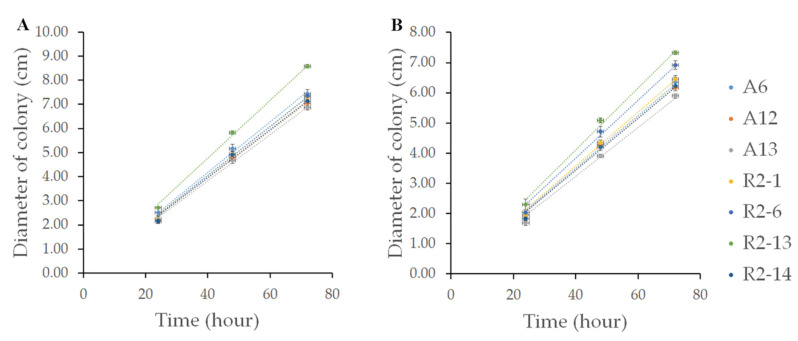
The radial growth rate of nematophagous strains. Mycelium plugs of 0.5 cm in diameter were inoculated on PDA (**A**) or MMV (**B**) plates at 27 °C for 72 h in constant white light. Radial growth was measured in triplicate using the Image J software (version 1.52a). The plots show the average ± the standard deviation of three independent experiments.

**Figure 3 pathogens-10-00815-f003:**
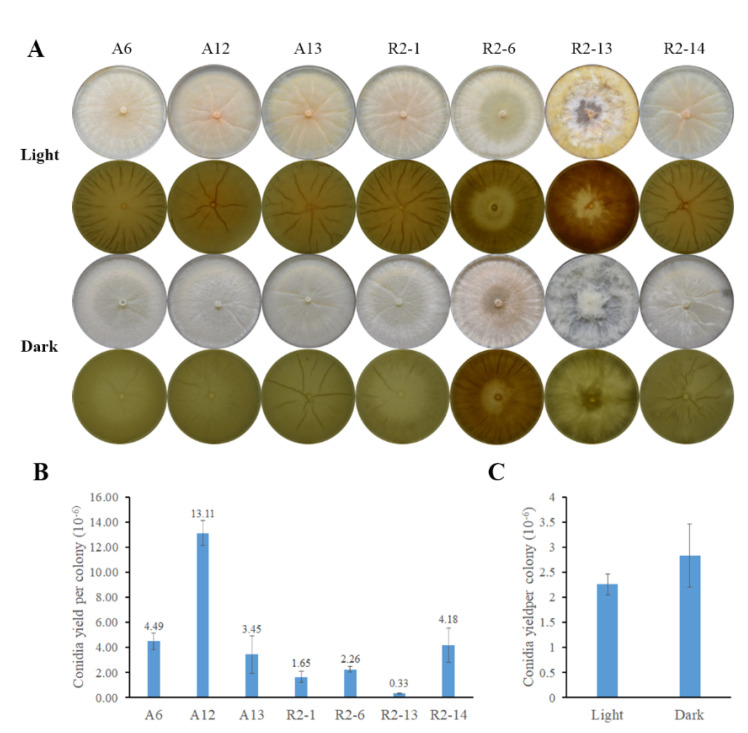
Conidial production stimulated by light in nematode-trapping isolates. (**A**) Colonies of the indicated strains were cultivated on PDA plates at 27 °C for 10 days under constant white light or darkness. Pictures were taken on a black background (first and third row) and white light transilluminator (second and fourth row). (**B**) The yield of conidia under white light. (**C**) Conidia production in the R2-6 strain comparing light and darkness conditions. The averages of the values ± the standard deviation of three independent experiments.

**Figure 4 pathogens-10-00815-f004:**
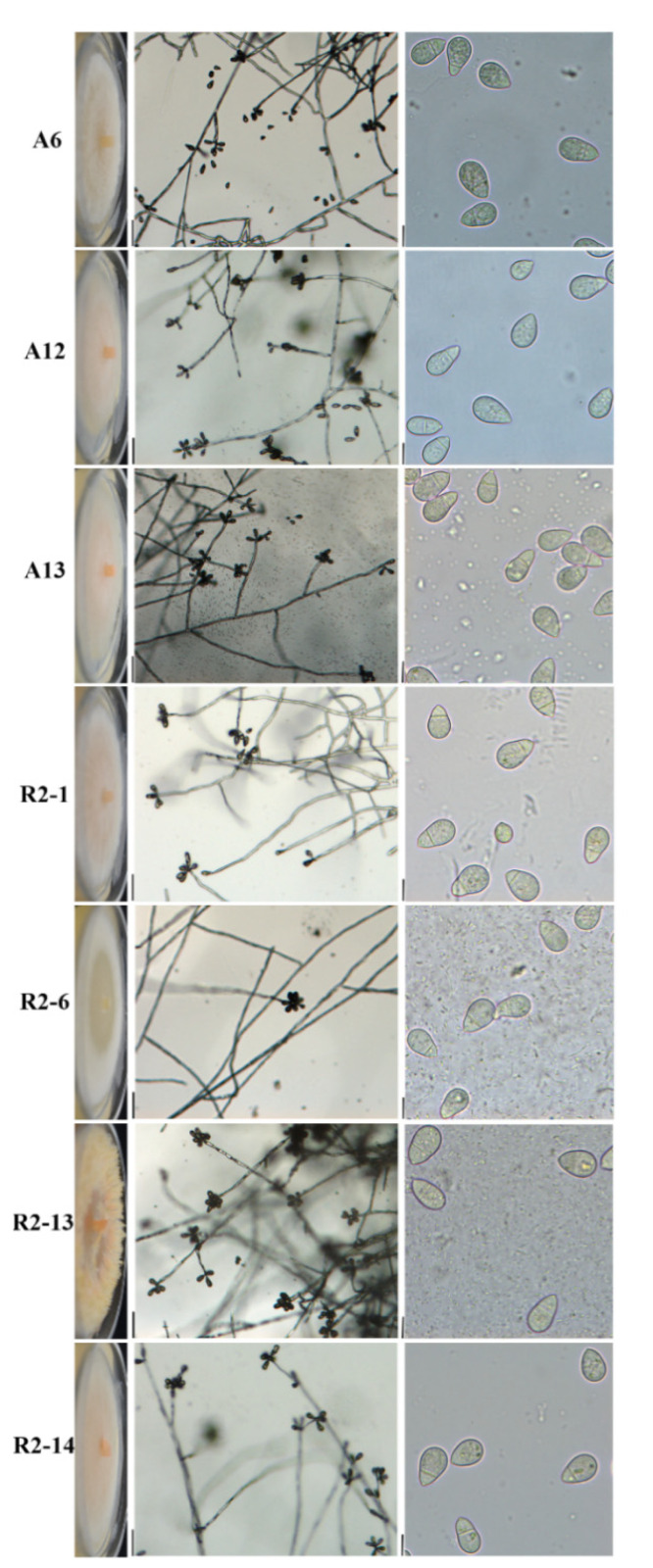
Morphology of colony and reproductive structures in nematode-trapping isolates. **Left** column, colonial morphology was capture from strains grown on PDA plates for 4 days at 27 °C. **Middle** column, 1 cm^3^ PDA blocks were inoculated with indicated strains, incubated at 27 °C for 7 days and conidiophores were analyzed at 10× magnification. The vertical line on the right side is 50 μm long. **Right** column, conidia collected from colonies grown on PDA at 27 °C for 10 days, as indicated in Figure 2, was analyzed at 40× magnification. The vertical line on the right side is 10 μm long.

**Figure 5 pathogens-10-00815-f005:**
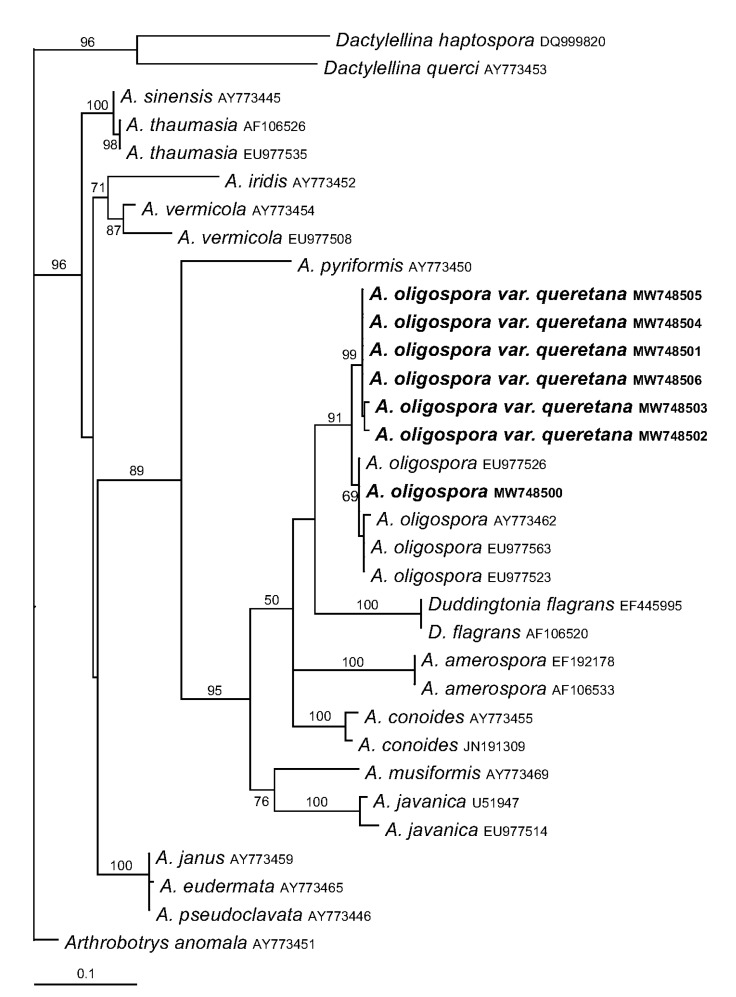
Maximum likelihood phylogenetic tree based on ITS marker. The tree was built using sequences indicated in Table 1. The bootstrap of maximum likelihood is indicated at the branches, only supports above 50 are shown.

**Figure 6 pathogens-10-00815-f006:**
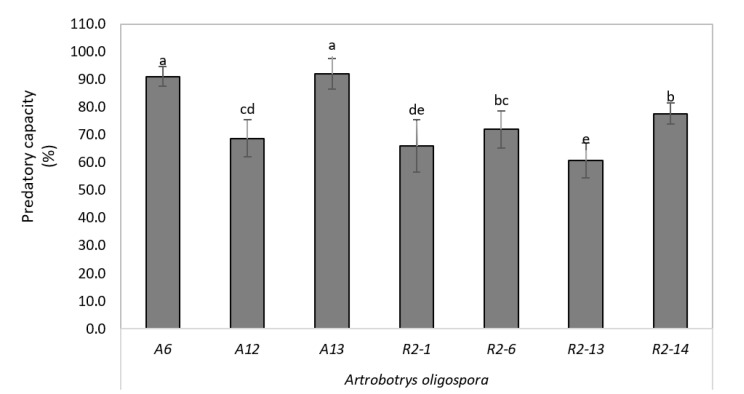
Predatory capacity in *Arthrobotrys oligospora* strains against *H. contortus* in vitro. Letters above bars represent the results of the statistical analysis; there was not a statistically significant difference in predatory capacity between strains sharing a given letter.

**Table 1 pathogens-10-00815-t001:** Reference and isolated nematophagous strains used in this study. ND, not declared by authors.

Species	Source	Region	Gene Bank	Reference
*Dactylellina haptospora*	ND	ND	DQ999820	[6]
*Dactylellina querci*	ND	ND	AY773453	[6]
*Arthrobotrys anomala*	Soil	China	AY773451	[6]
*A. amerospora*	ND	ND	EF192178	[21]
*A. amerospora*	ND	Germany	AF106533	[21]
*A. conoides*	Soil	Brazil, Paraná	JN191309	[21]
*A. conoides*	Soil	China	AY773455	[6]
*A. eudermata*	Soil	China	AY773465	[6]
*Duddingtonia flagrans*	Jardine soil	Germany, Berlín-Dahlem	AF106520	[21]
*D. flagrans*	Vineyard soil	USA, California	EF445995	[21]
*A. iridis*	Soil	China	AY773452	[6]
*A. janus*	Soil	China	AY773459	[6]
*A. javanica*	Soil	Indonesia, Java Island	U51947	[21]
*A. javanica*	Soil	ND	EU977514	[21]
*A. musiformis*	Soil	China	AY773469	[6]
*A. oligospora*	Soil	China	AY773462	[6]
*A. oligospora*	Decaying wood	ND	EU977526	[21]
*A. oligospora*	Decaying wood	ND	EU977563	[21]
*A. oligospora*	dung	ND	EU977523	[21]
*A. pseudoclavata*	Soil	China	AY773446	[6]
*A. pyriformis*	Soil	China	AY773450	[6]
*A. sinensis*	Soil	China	AY773445	[6]
*A. thaumasia*	Decaying wood	ND	EU977535	[21]
*A. thaumasia*	Cold greenhouse	Germany, Berlín-Dahlem	AF106526	[21]
*A. vermicola*	Soil	China	AY773454	[6]
*A. vermicola*	Decaying leaves	ND	EU977508	[21]
*A. oligospora var. queretana* A6	Soil underneath a tree	México, Querétaro, El Marqués	MW748504	This work
*A. oligospora var. queretana* A12	Compost	México, Querétaro, El Marqués	MW748505	This work
*A. oligospora* A13	Ovine feces	México, Querétaro, El Marqués	MW748506	This work
*A. oligospora var. queretana* R2-1	Donkey feces	México, Querétaro, Jalpan	MW748501	This work
*A. oligospora var. queretana* R2-6	Ovine feces	México, Querétaro, Jalpan	MW748502	This work
*A. oligospora var. queretana* R2-13	Bovine feces	México, Querétaro, Jalpan	MW748500	This work
*A. oligospora var. queretana* R2-14	Compost	México, Querétaro, Jalpan	MW748503	This work

**Table 2 pathogens-10-00815-t002:** Measurement of *Arthrobotrys* two-celled conidia. Two-celled conidia from colonies grown on PDA for 10 days were measured as indicated in Appendix A. BC-S, Basal Cell–Septum; S-DC, Septum–Distal Cell, and in parentheses are indicated the averages. “***n***” in parentheses indicates the counted conidia population.

Strain	Conidia Length (μm)	Septum (μm)	BC-S (μm)	S-DC (μm)	S-DC/BC-S
A6 (*n* = 132)	15.2–24.4 (18.4)	5.3–11.3 (8.4)	4.0–11.4 (6.7)	8.4–15.3 (11.7)	1.74
A12 (*n* = 100)	15.6–21.6 (18.6)	6.7–10.8 (8.0)	4.4–9.1 (6.7)	9.5–14.7 (11.9)	1.79
A13 (*n* = 105)	15.4–23.1 (18.8)	6.4–11.3 (8.6)	4.5–9.2 (7.0)	8.6–14.4 (11.9)	1.71
R2-1 (*n* = 96)	15.0–21.9 (18.5)	6.5–11.6 (9.0)	3.8–8.8 (6.7)	9.1–14.6 (11.9)	1.78
R2-6 (*n* = 99)	13.9–22.1 (17.5)	6.2–12.1 (8.4)	3.6–8.9 (6.4)	7.3–13.9 (11.1)	1.75
R2-13 (*n* = 100)	16.0–30.3 (21.4)	7.3–14.8 (8.2)	6.4–15.8 (8.9)	9.0–15.9 (12.4)	1.39
R2-14 (*n* = 100)	15.1–22.2 (18.2)	7.0–11.9 (9.0)	4.1–8.9 (6.4)	8.4–15.1 (11.8)	1.85

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
