# Peer review of "High Predatory Capacity of a Novel Arthrobotrys oligospora Variety on the Ovine Gastrointestinal Nematode Haemonchus contortus (Rhabditomorpha: Trichostrongylidae)"

_pathogens, 2021, doi:10.3390/pathogens10070815_

Round 1

Reviewer 1 Report

This manuscript describes the isolation of several strains of the much studied nematophagous fungus Arthrobotrys oligospora in the Mexican State of Querétaro, and their predatory activity against Haemonchus contortus larvae in vitro. The novel element in the work described seems to be the geographical location of the isolates, as similar work has already been published and should be cited (see “In vitro predatory activity of nematophagous fungi isolated from water buffalo feces and from soil in the Mexican southeastern, Ojeda-Robertos et al., 2019)( https://doi.org/10.1590/S1984-29612019011). Beyond that, listed below are some points the authors could consider.

  1. Introduction, l. 44-46. The text “nematophagous fungi play a key role as natural enemies of parasitic nematodes of plants and animals”, should be labelled as a “likely” or perhaps “probable” situation, unless there exist studies describing nematode abundance in comparable environments with and without nematophagous fungi, in which case they should be cited.
  2. Introduction, l. 68: References 12 and 13 do not seem to be indexed in either PubMed or Scopus, so it is difficult to judge the relevance of the statement they intend to support. Mendoza de Gives et al. (2018) is a more accessible reference for the argument about cattle (https://doi.org/10.1016/j.vetpar.2018.10.001).
  3. Materials & Methods, l. 119: Units for measurements of light exposure are not micromolar.
  4. Materials & Methods, l. 143: Powders are typically not measured using units of volume.
  5. Materials & Methods, l. 241-243: The text states that “colony color was directly associated with conidia production, suggesting that pigment production is associated with asexual reproduction in nematode-trapping fungi.” But Figure 3A seems to show that the R2-6 & R2-13 strains are the most pigmented, yet they have the lowest spore yield. Perhaps the apparent discrepancy could be clarified in the main text or in the figure legend.
  6. Results 3.5, l. 291-296: The authors claim the discovery of a previously undescribed variant of oligospora, without discussing their possible relationship to previously published Mexican variants (Ojeda-Robertos et al., 2019)( https://doi.org/10.1590/S1984-29612019011).
  7. Results 3.5, l. 305: To avoid readers being misled, it would be useful to clarify, here or in the Discussion, that predatory activity of fungal isolates was measured in vitro only, while in soil or the animal intestinal tract their activity could be different, as it is known to be influenced by nutrient availability and other external cues.
  8. Figure 6: Labelling bars with different letters to indicate statistically significant differences is very confusing for the reader.

Author Response

Manuscript ID:  pathogens-1233083

Title: High predatory capacity of a novel Arthrobotrys oligospora variety on the ovine gastrointestinal nematode Haemonchus contortus (Rhabditomorpha: Trichostron-gylidae).

We thank to the reviewers for all the assertive comments, all of them helped us improve the quality of the manuscript. Below, we address each of your questions and comments:

Reviewer #1:

  1. Introduction, l. 44-46. The text “nematophagous fungi play a key role as natural enemies of parasitic nematodes of plants and animals”, should be labelled as a “likely” or perhaps “probable” situation, unless there exist studies describing nematode abundance in comparable environments with and without nematophagous fungi, in which case they should be cited.

Response: Thank you for your comment. We have modified the sentence, according to your suggestion as “…nematophagous fungi likely play a role as natural enemies of…”.

  1. Introduction, l. 68: References 12 and 13 do not seem to be indexed in either PubMed or Scopus, so it is difficult to judge the relevance of the statement they intend to support. Mendoza de Gives et al. (2018) is a more accessible reference for the argument about cattle (https://doi.org/10.1016/j.vetpar.2018.10.001).

Response: Thank you for your suggestion, which helps us improve the quality of our amnsucript. We substituted references 12-13 for the following: 

  1. Ojeda-Robertos, N.F.; Aguilar-Marcelino, L.; Olmedo-Juárez, A.; Luna-Palomera, C.; Peralta-Torres, J.A.; López-Arellano, M.E.; Mendoza-de-Gives, P. In vitro predatory activity of nematophagous fungi isolated from wa-ter buffalo feces and from soil in the Mexican southeastern. Braz. J. Vet. Parasitol., Jaboticabal, 2019, 28, 314-319. Doi: https://doi.org/10.1590/S1984-29612019011.
  2. Mendoza-de-Gives, P.; López-Arellano, M.E.; Aguilar-Marcelino, L.; Olazarán-Henkins, S.; Reyes-Guerrero, D.; Ra-mírez-Várgas, G.; Vega-Murillo, V.E. The nematophagous fungus Duddingtonia flagrans reduces the gastrointesti-nal parasitic nematode larvae population in faeces of orally treated calves maintained under tropical condi-tions—Dose/response assessment. Veterinary Parasitology 2018, 263, 66–72. https://doi.org/10.1016/j.vetpar.2018.10.001.

  1. Materials & Methods, l. 119: Units for measurements of light exposure are not micromolar.

Response:

We measured the white-light quantity with a Portable Lux Meter (HI97500) by Hanna Instruments and the result (3.6 klux) was converted to µmol m-2.

Conversion was based on the appendix: Light calibrations and radiometric units. In: Phycomyces, pp. 375–380, Cerdá-Olmedo, E., Lipson, E.D., eds. Cold Spring Harbor Laboratory, New York.

We have reported the following information: The reference intensity of 0.1 W m-2 of broadband blue light corresponds to an incident energy fluence rate of 10 µW cm-2, or 10-1 W m-2; to a photon fluence rate of 2.3 x 1013 quanta cm-2 s-1, 2.3 x 1017 quanta m-2 s-1, or 3.8 x 10-7 mol m-2 s-1 (note: 1 einstein = 1 mol of photons); and to a total flux of 2500 meter-candles or 250 foot-candles (fc; 1 fc= 10.764 lux).

If 1 fc = 10.764 lux

3600 luxes = 334.448 fc

If 250 fc = 3.8 x 10-7 mol m-2 s-1

334.448 fc = 5.0836 x 10-7 mol m-2 s-1

If 1 mol = 1 x 106 µmol

5.08 x 10-7 mol m-2 s-1 = 0.508 µmol m-2 s-1

We have carried out several fungal photobiology studies, and the fluence rate is reported using this units:

  1. Calcáneo-Hernández G, Rojas-Espinosa E, Landeros-Jaime F, Cervantes-Chávez JA, Esquivel-Naranjo EU. An efficient transformation system for Trichoderma atroviride using the pyr4 gene as a selectable marker. Braz J Microbiol. 2020 Dec;51(4):1631-1643. doi: 10.1007/s42770-020-00329-7. Epub 2020 Jul 6.
  2. Esquivel-Naranjo EU, García-Esquivel M, Medina-Castellanos E, Correa-Pérez VA, Parra-Arriaga JL, Landeros-Jaime F, Cervantes-Chávez JA, Herrera-Estrella A. A Trichoderma atroviride stress-activated MAPK pathway integrates stress and light signals. Mol Microbiol. 2016 Jun;100(5):860-76. doi: 10.1111/mmi.13355. Epub 2016 Apr 20.
  3. Castellanos F, Schmoll M, Martínez P, Tisch D, Kubicek CP, Herrera-Estrella A, Esquivel-Naranjo EU. Crucial factors of the light perception machinery and their impact on growth and cellulase gene transcription in Trichoderma reesei. Fungal Genet Biol. 2010 May;47(5):468-76. doi: 10.1016/j.fgb.2010.02.001. Epub 2010 Feb 6.
  4. Esquivel-Naranjo EU, Herrera-Estrella A. Enhanced responsiveness and sensitivity to blue light by blr-2 overexpression in Trichoderma atroviride. Microbiology (Reading). 2007 Nov;153(Pt 11):3909-3922. doi: 10.1099/mic.0.2007/007302-0.

  1. Materials & Methods, l. 143: Powders are typically not measured using units of volume.

Response: the reviewer´s concern is right, but to avoid defrosting the samples, it was easier and faster to add the same volume of powered mycelium with the aim to increase the DNA quality. We modified the sentence to “The equivalent to five-hundred microliters of powder mycelia were added…”

  1. Materials & Methods, l. 241-243: The text states that “colony color was directly associated with conidia production, suggesting that pigment production is associated with asexual reproduction in nematode-trapping fungi.” But Figure 3A seems to show that the R2-6 & R2-13 strains are the most pigmented, yet they have the lowest spore yield. Perhaps the apparent discrepancy could be clarified in the main text or in the figure legend.

Response: Thank you for your comment, it was clarified and corrected. The sentence reads now “Furthermore, although colony pigmentation and asexual reproduction was photoinduced, there was not relationship between colony color intensity and conidia yield, suggesting that mycelium and reproductive structures could be pigmented in nematode-trapping fungi. Our results suggest that those isolates correspond to at least two different kinds of nematode-traping fungi.”

  1. Results 3.5, l. 291-296: The authors claim the discovery of a previously undescribed variant of oligospora, without discussing their possible relationship to previously published Mexican variants (Ojeda-Robertos et al., 2019) (https://doi.org/10.1590/S1984-29612019011).

Response: We appreciate your comment to this section. We revised the suggested paper. Although the work published the identification of nematophagous fungi, the authors only use classical taxonomy, without documented images of reproductive structures, form, sizes, and they did not use molecular taxonomy using the entire internal transcribed spacer region (ITS1-5.8S-ITS2), therefore we cannot compare our work versus the work suggested. In fact, it is really complicated get species and variety by classical taxonomy.

  1. Results 3.5, l. 305: To avoid readers being misled, it would be useful to clarify, here or in the Discussion, that predatory activity of fungal isolates was measured in vitro only, while in soil or the animal intestinal tract their activity could be different, as it is known to be influenced by nutrient availability and other external cues.

Response: Thank you for your comments. To avoid confusion, we have specified in section 3.6 that the predatory capacity of H. contortus was evaluated in vitro: “3.6. High depredatory capacity of nematophagous native against H. contortus in vitro

The nematode H. contortus is the main sheep gastrointestinal nematode causing large losses in production systems. To explore the potential application, we analyze the depredatory capacity in nematode-trapping isolates against H. contortus in vitro. As well as in the Figure 6 legend: “Predatory capacity in Arthrobotrys oligospora strains against H. contortus in vitro.”

  1. Figure 6: Labelling bars with different letters to indicate statistically significant differences is very confusing for the reader.

Response: Thank you for your comment. The statistical analysis indicated differences between the different groups, so if two or more groups have the same letter, it means there is no statistical difference. We have clarified that on the figure legend as follows: “Literals above bars represent the results of the statistical analysis, where a different letter indicates a significant statistical difference among strains.”

Reviewer 2 Report

The paper is based in the novelty on look for new alternative methods for controlling these nematodes using soil nematode nematophagous fungi Arthrobotys oligospora. Results are very interesting and the experiments performed follow canonical and comprensible process. They have demonstrated there are at least two groups of Arthrobotrys oligospora in Querétaro from a phylogenetic point of view and tested it from predatory experiments. In addition the paper provides interesting phenotypic characteristic for what is propossed as Arthrobotys oligospora vr. queretana.

Suggestions:

a) Change  the title such as "High predatory capacity of a novel Arthrobotrys oligospora variety on ovine gastrointestinal nematode Haemonchus contortus (Rhabditomorpha: Trichostrongylidae)

b) page 2, modify sentence in line 84 after ".....underneath trees." such as: ".....underneath trees. In total 55 samples were taken and stored in..."

c) Table 1.

Clarify what is ND (not declared?)

Is Amazcala El marques?, please clarify (also in paragraph from line 80-87

d) Fig. 1

Please, include notation of own isolates from Table 1, such as all the isolates from A. oligospora vr. queretana  group be labeled also with  (A12), (A6), (R2-1), (A13), (R2-14) and (R2-6) while A. oligospora MW748500 be also labeled with (R2-13)

Author Response

Manuscript ID:  pathogens-1233083

Title: High predatory capacity of a novel Arthrobotrys oligospora variety on the ovine gastrointestinal nematode Haemonchus contortus (Rhabditomorpha: Trichostron-gylidae).

We thank to the reviewers for all the assertive comments, all of them helped us improve the quality of the manuscript. Below, we address each of your questions and comments:

Reviewer #2:

  1. Change the title such as "High predatory capacity of a novel Arthrobotrys oligospora variety on ovine gastrointestinal nematode Haemonchus contortus (Rhabditomorpha: Trichostrongylidae)

Response:

We agree with your suggestion, and we modified the title of the manuscript: “High predatory capacity of a novel Arthrobotrys oligo-spora variety on the ovine gastrointestinal nematode Haemonchus contortus (Rhabditomorpha: Trichostron-gylidae).

  1. b) page 2, modify sentence in line 84 after ".....underneath trees." such as: ".....underneath trees. In total 55 samples were taken and stored in..."

Response: Thank you for your suggestion, which helps us improve the quality of our manuscript. The modified the sentence this way: “and underneath trees. In total 55 samples were collected and stored in polyethylene bags…”

  1. c) Table 1.

Clarify what is ND (not declared?)

Response. Thank you for your comment. We have modified the Table 1 title: “Table 1. Reference and isolated nematophagous strains used in this study. ND, not declared by authors.

Is Amazcala El marques?, please clarify (also in paragraph from line 80-87

Response, for clarity, we have modified section 2.1 in the following way: “The study was carried out from June 2018 through September 2018, in three locations in the Mexican state of Querétaro: one in El Marqués (2042’37.5”N, 10015’24.6”W) and two in Jalpan de Serra (2111’31.7”N, 9927’23.0” W) and  (2109159.1”N, 9921’01.7”W).

  1. d) Fig. 1

Please, include notation of own isolates from Table 1, such as all the isolates from A. oligospora vr. queretana  group be labeled also with  (A12), (A6), (R2-1), (A13), (R2-14) and (R2-6) while A. oligospora MW748500 be also labeled with (R2-13)

Response. Thank you for your valuable suggestion. We modified Table 1 accordingly.

Reviewer 3 Report

The present manuscript presents research work on several A. oligospora isolates that differ in predatory activity, morphology and faintly in their ITS sequences from known reference strains.

However, several points need a revision before this manuscript can be accepted for publication:

  • Genus classification of Arthrobotrys flagrans. This is an old and outdated designation for Duddingtonia flagrans (Duddingtonia flagrans (Dudd.) R.C. Cooke, Transactions of the British Mycological Society 53 (2): 316 (1969) [MB#330246]. Please change this in the entire manuscript incl. Tab1 and Fig. 5. Refer to the mycobank for the actual designations of genus names: https://www.mycobank.org/page/Simple%20names%20search

Further, there are many flaws in English style and partially also in clarity of the description of some parts in the manuscript. Below some mistakes but certainly not all, since me neither I am a native English speaker. I recommend to let the manuscript further corrected by a native English speaker.

Line 22: includes (instead of include)

Line 27: greater than 58%: in relation to what? What is the reference? Did it predate and kill 58% of the nematodes but in which setup? Indicate that this was done on water agar plates.

Line 30: is promissory the right word? Shouldn’t it be promising?

Line 54: here it should be rather stated “as a biocontrol alternative for use of anthelmintic for the control of infectious larvae”

Line 56: resistant spores not resistance spores

Line 62: livestock feed not food

Line 63: elimination is the wrong word. Better excretion or faeces deposition

Line 76: biological control agents or biocontrol agents (BCA)

Line 82/83: samples were collected from …. excrements and from the soil underneath the excrements and trees. Why trees? Because of leaf litter decomposed by nematodes?

Line 96b: use water agar (WA) instead of agar water (AW)

Line 159/160: The final incubation at 72°C serves as final elongation to complement any fragments not processed yet to the final length and not “to stop the reaction”.

Line 160: not the amplifications are analyzed but the amplification products

Line 162: how the sequences were trimmed? Please indicate length or other criteria

Line 175: please be more detailed: you are inoculating water agar plates with plugs (5 mm). Please indicate where the plugs are coming from (I guess from a PDA culture, but of what age and were the plugs taken in the center (old mycelium) or at the periphery of the growing culture?

Line 176: was the number of trapping structure quantified per plate?

Line 182: the volume of the tube was reduced (better than adjusted). How was it done? By taking off the liquid with a pipette or by decanting?

Line 284: Table 2: S-AC/BC-S, correct typing error

Line 304: I would say: depredatory capacity of isolates (not in isolates)

Line 315: usually the term biocontrol agent is used (BCA)

What I am missing in the discussion:

Why are they promising as BCA of helminthic parasites as you suggested in the last sentence of the abstract?

What can be the application and importance of A. oligospora isolates as BCA in the real world? Application as a conidia spray on meadows? Please discuss such an approach in comparison with approaches using Duddingtonia flagrans applied as livestock feed additive.

Author Response

Manuscript ID:  pathogens-1233083

Title: High predatory capacity of a novel Arthrobotrys oligospora variety on the ovine gastrointestinal nematode Haemonchus contortus (Rhabditomorpha: Trichostron-gylidae).

We thank to the reviewers for all the assertive comments, all of them helped us improve the quality of the manuscript. Below, we address each of your questions and comments:

Reviewer #3:

The present manuscript presents research work on several A. oligospora isolates that differ in predatory activity, morphology and faintly in their ITS sequences from known reference strains.

However, several points need a revision before this manuscript can be accepted for publication:

Genus classification of Arthrobotrys flagrans. This is an old and outdated designation for Duddingtonia flagrans (Duddingtonia flagrans (Dudd.) R.C. Cooke, Transactions of the British Mycological Society 53 (2): 316 (1969) [MB#330246]. Please change this in the entire manuscript incl. Tab1 and Fig. 5. Refer to the mycobank for the actual designations of genus names: https://www.mycobank.org/page/Simple%20names%20search

Response: We appreciate your revision, which helps improve the quality of our manuscript. We have substituted the name Arthrobotrys flagrans for Duddingtonia flagrans in both, Table 1 and Figure 5.

Further, there are many flaws in English style and partially also in clarity of the description of some parts in the manuscript. Below some mistakes but certainly not all, since me neither I am a native English speaker. I recommend to let the manuscript further corrected by a native English speaker.

Response: We appreciate your comment. The manuscript was edited for proper English language, grammar, punctuation, spelling, and overall style by the American Manuscript Editors company. We include the certificate.

Line 22: includes (instead of include)

Response: thank you, we have corrected the mistake. “The nematode-trapping fungi were classified into two groups; the first group includes the R2-13 strain…”

Line 27: greater than 58%: in relation to what? What is the reference? Did it predate and kill 58% of the nematodes but in which setup? Indicate that this was done on water agar plates.

Response: We appreciate your question. We obtained this percentage of reduction on predatory capacity using a formula described in the methodology section, and it is obtained comparing the results with those of the control group. Therefore, for clarity, we modified the sentence as follows: “In all the strains, the predatory capacity against the sheep gastrointestinal nematode Haemonchus contortus was greater than 58% compared with the control group.”

Line 30: is promissory the right word? Shouldn’t it be promising?

Response: according to the context of the sentence, and in agreement with the language Editor, promissory is the right word. Thank you.

Line 54: here it should be rather stated “as a biocontrol alternative for use of anthelmintic for the control of infectious larvae”

Response: We appreciate your suggestion. The language Editor suggested the following edited sentence: “…as a biocontrol alternative for the use of anthelmintics in infective larvae of parasites in domestic animals.”

Line 56: resistant spores not resistance spores

Response: thank you for the notice, it has been corrected.

Line 62: livestock feed not food

Response. Thank you for your correction, which helps improve the quality of the manuscript. The sentence was modified as follows: “…that can be added to livestock feed as additives to control parasites…”

Line 63: elimination is the wrong word. Better excretion or faeces deposition

Response: Thank you, the sentence was modified as follows: “…the gastrointestinal tract and, when excreted into the environment…”

Line 76: biological control agents or biocontrol agents (BCA)

Response. We have modified the sentence for clarity as follows: “…and as a first approach that can be used as biological control agents for infective larvae of ruminant gastrointestinal nematodes.”

Line 82/83: samples were collected from …. excrements and from the soil underneath the excrements and trees.

Response: The edited sentence is as follows: “Samples of feces from sheep, cattle and horses were collected from the soil in depth ranges of 0 to 5 cm and in compost, pasture soil and under trees.”

Why trees? Because of leaf litter decomposed by nematodes?

Response: Yes.

Line 96b: use water agar (WA) instead of agar water (AW)

Response: Thank you, we modified the term to water-agar (WA).

Line 159/160: The final incubation at 72°C serves as final elongation to complement any fragments not processed yet to the final length and not “to stop the reaction”.

Response: We have edited the sentence for clarity like this: “The following thermocycling conditions were carried out: an initial denaturation step at 95 ° C for 3 min followed by 40 cycles, each containing a denaturation step of 95 ° C for 30 s, an alignment step at 58 ° C for 30 s, and an extension step at 72 ° C for 45 s. Reactions were stopped after a final extension step at 72 ° C for 5 min.”

Line 160: not the amplifications are analyzed but the amplification products

Response: thank you, we have modified the sentence accordingly: “The amplicons were analyzed by agarose gel electrophoresis at 1%...”

Line 162: how the sequences were trimmed? Please indicate length or other criteria

Response: We analyzed the sequences using Chromas software to select the sequences with high quality, with well-defined peaks in the electropherograms. In the manuscript we edited the sentence for clarity as follows: “The sequences were trimmed to select those with high quality, with well-defined peaks in the electropherograms using Chromas software…”

Line 175: please be more detailed: you are inoculating water agar plates with plugs (5 mm). Please indicate where the plugs are coming from (I guess from a PDA culture, but of what age and were the plugs taken in the center (old mycelium) or at the periphery of the growing culture?

Response: Thank you for your comment. The sentence was edited for clarity: “Each fungus was cultured in water-agar Petri dishes (5 mm plugs) for 3 weeks at room temperature (25-28 ° C), from 7-day-old pure colonies grown on PDA plates and obtained from the periphery of the culture.”

Line 176: was the number of trapping structure quantified per plate?

Response: The number of capture structures per plate was not quantified, the objective of the work was to analyze the action of the fungus against the parasite and to determine the total percentage of capture.

Line 182: the volume of the tube was reduced (better than adjusted). How was it done? By taking off the liquid with a pipette or by decanting?

Response: Thank you for your comment, the sentence was edited as follows: “The volume of the tubes was reduced to 2 ml by means of a pipette, where the larvae were found.”

Line 284: Table 2: S-AC/BC-S, correct typing error

Response: we corrected the typo, thank you!

Line 304: I would say: depredatory capacity of isolates (not in isolates)

Response. Thank you for your comment, we edited the sentences as follows: “To explore the potential application, we analyze the depredatory capacity of nematode-trapping isolates against H. contortus in vitro.”

Line 315: usually the term biocontrol agent is used (BCA)

Response: thank you, we modified the term: “Nematophagous fungi are potential biocontrol agents…”

What I am missing in the discussion:

Why are they promising as BCA of helminthic parasites as you suggested in the last sentence of the abstract?

What can be the application and importance of A. oligospora isolates as BCA in the real world? Application as a conidia spray on meadows? Please discuss such an approach in comparison with approaches using Duddingtonia flagrans applied as livestock feed additive.

Response: We understand your point, we added the following ideas in Discussion section: “A. oligospora is the nematophagous fungus that is found in most parts of the world, which suggests that it is the one that best adapts to different environmental conditions, not being the case of D. flagrans, which needs certain climatic characteristics for its development and nematophagous action. On the other hand, it is intended to be able to carry out tests in the future of the ways of applying it, for example: to elaborate concentrate feed that contain these spores and / or conidia. In the same way, formulated granules containing the fungal material can be prepared. This research work opens the door to future research on biological control using nematophagous fungi, characterizing new isolates that will also be explored for better action and efficacy.”

Round 2

Reviewer 1 Report

The authors have satisfactorily addressed most issues raised in the review. However, I still think the new reference 12 should also be mentioned in the Introduction, and keep needing a long time to understand the differences in statistical significance in Fig. 6. I would suggest, for the authors’ consideration, an alternative wording such as “there was not a statistically significant difference in predatory capacity between strains sharing a given letter”.

Author Response

Thank you for your prompt response.

Que have adressed your coments:

a) the reference No. 12 was already included in the Introduction of the text line 64.

b) we modified the estructure of the sentence on the Figure 6 leyend, thank you.
